# Effect of Carotid Stenosis Severity on Patterns of Brain Activity in Patients after Cardiac Surgery

**Irina Tarasova \***, **Olga Trubnikova, Darya Kupriyanova, Irina Kukhareva, Irina Syrova, Anastasia Sosnina, Olga Maleva** and **Olga Barbarash**

Department of Clinical Cardiology, Research Institute for Complex Issues of Cardiovascular Diseases, 6, Sosnoviy Blvd., 650002 Kemerovo, Russia

\* Correspondence: iriz78@mail.ru

**Abstract:** Background: The negative effects of high-grade carotid stenosis on the brain are widely known. However, there are still insufficient data on the brain state in patients with small carotid stenosis and after isolated or combined coronary and carotid surgery. This EEG-based study aimed to analyze the effect of carotid stenosis severity on associated brain activity changes and the neurophysiological test results in patients undergoing coronary artery bypass grafting (CABG) with or without carotid endarterectomy (CEA). Methods: One hundred and forty cardiac surgery patients underwent a clinical and neuropsychological examination and a multichannel EEG before surgery and 7–10 days after surgery. Results: The patients with CA stenoses of less than 50% demonstrated higher values of theta2- and alpha-rhythm power compared to the patients without CA stenoses both before and after CABG. In addition, the patients who underwent right-sided CABG+CEA had generalized EEG "slowdown" compared with isolated CABG and left-sided CABG+CEA patients. Conclusions: The on-pump cardiac surgery accompanied by specific re-arrangements of frequency–spatial patterns of electrical brain activity are dependent on the degree of carotid stenoses. The information obtained can be used to optimize the process of preoperative and postoperative management, as well as the search for neuroprotection and safe surgical strategies for this category of patients.

**Keywords:** carotid stenosis; brain electrical activity; EEG; postoperative cognitive dysfunction; coronary artery bypass grafting; carotid endarterectomy





## 1. Introduction

According to the World Health Organization, cardiovascular diseases (CVD), mainly associated with atherosclerosis, are the leading causes of death worldwide, including in Russia [1]. The Siberian region shows less favorable CVD epidemiology. Various climatic and ecological conditions of the region contribute to the high prevalence of this pathology [2]. Atherosclerosis often affects multiple vascular basins simultaneously. Significant atherosclerotic lesions of several vascular basins determine the severity of the disease, making it difficult to choose the optimal treatment strategy and calling into question the positivity of the prognosis, in particular, coronary artery disease.

The global population of elderly people has been increasing every year, and the ageing of the population has posed new and complex challenges for health professionals not only to increase life expectancy but also to maintain its quality. A high standard of quality of life cannot be reached without preserving a person's intellectual functions. It is known that with age, cognitive functions diminish, and cognitive impairment (CI) develops in the form of memory loss, attention and executive impairment, etc. [3–5].

Cognitive disorders associated with cerebral and coronary atherosclerosis (vascular CI) are widespread among older persons and are more severe than age-related cognitive changes [6–8]. Previous studies have revealed significant interactions between cognitive disorders developing in the elderly and senile age, atherosclerotic changes in cerebral

vessels, and accompanying disorders of cerebral blood flow [9,10]. There is evidence that age-related structural and functional changes in arteries, arterioles and capillaries lead to dysregulation of cerebral blood flow and ischemia, leading to disruption of the blood–brain barrier. Additionally, metabolic disorders are developed with reduced delivery of energy substrates to neurons and excretion of by-products of the protein breakdown, increasing neuroinflammation and paracrine regulation dysfunction [11,12]. It is suggested that the atherosclerotic remodeling of the brain vessels can lead to an accelerated progression of brain dysfunction [11]. In this case, carotid artery (CA) stenosis is one of the factors affecting self-regulation of brain perfusion [13]. It has been found that patients with vascular CI often show a decrease in blood flow velocity in the cerebral cortex, especially in the frontal and parietal regions [14,15]. These brain regions are known to be the watersheds of the blood supply, at the boundaries between the vascular pools [16–18]. These zones are more disadvantaged than any other brain region in the case of systolic and/or diastolic dysfunction of the left ventricle, valvular pathology and atrial fibrillation accompanying cardiovascular pathology, as well as during cardiac surgery [11,19].

There is a wide variety of epidemiological and clinical data on vascular CI, but only a few studies have examined changes in the neurophysiological parameters of cardiac surgery patients [20,21]. At the same time, early manifestations of vascular and postoperative CI are subclinical and are detected only using an extended neurophysiological examination. In this regard, careful attention should be paid to the identification of objective and sensitive criteria for early diagnosis of CI in cardiac surgery patients. It is generally accepted that the electroencephalogram (EEG) rhythms reflect the activity of the neural network to be placed under recording electrode [22,23]. As a consequence, the changes in EEG rhythms may be early indicators of structural and functional abnormalities in neural networks associated with vascular and postoperative CI.

Previous studies have shown that the frequency–spatial pattern of brain electrical activity in patients with vascular CI has specific features [8,24,25]. The association between poststroke alpha slowing and CI, which may be mediated by attentional dysfunction, was revealed [24]. Al-Qazzaz et al. [25] studied the discriminatory characteristics of patients with vascular CI and healthy individuals using non-linear EEG analysis methods. It was found that the degree of EEG irregularity and complexity was significantly lower in patients with vascular CI compared to control subjects. We previously showed that a theta activity increase in the frontal and occipital sites, as well as high theta/alpha ratios, may be considered as the earliest EEG markers of vascular cognitive disorders [8]. Moretti et al. proposed several promising EEG markers that could be important in the differential diagnosis of vascular and neurodegenerative CI. The alpha3/alpha2 and theta/gamma indices showed prognostic significance for the progression of the neurodegenerative type of CI [26–28]. The changes in the electrical activity of neurons in the post-stroke period proved to be promising in the search for prognostic markers of clinical recovery in patients with ischemic brain damage. A study by Zappasodi et al. found that a bilateral increase in low-frequency activity and a decrease in hemispheric asymmetry in the acute phase of a unilateral stroke in the middle cerebral artery basin predicts a worse functional outcome in the future [29].

However, there is still insufficient information on the modification of the brain electrical activity in cardiac surgery patients. Cardiac surgery has been shown to be associated with local or diffuse brain damage [21,30–32]. It is assumed that chronic cerebral ischemia in patients with cardiovascular diseases, as well as episodes of acute ischemia that occur during on-pump cardiac surgery, can contribute to specific changes in the brain's electrical activity. Our previous studies have shown that EEG patterns associated with coronary artery bypass grafting (CABG) have specific features, depending on the presence of pre-operative CI or cognitive decline in the early postoperative period [21,33]. We found that the presence of early POCD was accompanied by negative postoperative dynamics of EEG parameters with the increase in low-frequency activity. Skhirtladze-Dworschak et al. found that the occurrence of nonconvulsive status epilepticus after open cardiac surgery is associated with mitigating secondary brain injury [34].

Thus, recent studies have shown that the patterns of brain activity are associated with perioperative brain damage in cardiac surgery patients. However, the role of the severity of carotid stenosis in the development of the postoperative changes in brain activity and cognitive functions is uncertain. It has previously been shown that hemodynamically significant stenoses of CA (70–99%) can be a risk factor for brain damage during cardiac surgery [31,35]. However, little is known about the effects of small stenoses of CA (<50%) on the state of the brain in cardiac surgery patients. There have been several research studies into the negative effects of asymptomatic stenosis of CA on the state of the brain after cardiac surgery [33,35]. This has resulted in the perception that CA stenoses of less than 50% are hemodynamically insignificant. Therefore, this has led to insufficient attention being paid to preoperative management and intraoperative brain protection in patients with CA stenoses of less than 50%.

There are some data in the literature about the serious neurological complications (stroke, postoperative delirium, etc.) that occur in the group of patients with hemodynamically significant stenoses [36,37]. Research studies about the brain activity changes associated with postoperative cognitive decline in patients with stenoses of the coronary and carotid arteries are rare, especially after simultaneous cardiac surgery. It is important to note that the intraoperative episodes of brain ischemia during combined coronary and carotid revascularization does not necessarily lead to brain damage such as stroke. Meanwhile, less pronounced, diffuse ischemic brain damage may have a significantly higher frequency. Further, this may lead to a decline in cognitive functions and complicate the postoperative management of patients undergoing combined cardiac surgery.

In this paper, we will analyze the effect of carotid stenosis severity on associated EEG changes and the results of neurophysiological examination, including the frequency and structure of CI, in patients undergoing cardiac surgery (isolated CABG and combined CABG and carotid endarterectomy (CEA)) in the early postoperative period.

## 2. Materials and Methods

### 2.1. Subjects

This study was a prospective, observational cohort investigation. From a cohort of patients who underwent on-pump coronary surgery in the clinic of the Research Institute for Complex Issues of Cardiovascular Diseases, a sample of 140 subjects was selected. All of the patients met the study criteria and signed an informed consent form. The isolated CABG group consisted of 86 patients, 29 of whom had unilateral CA stenoses of less than 50%. The CABG+CEA group were divided into two groups: the group of left-sided CEA+CABG (n = 30) and the group of right-sided CEA+CABG (n = 24) (see Figure 1).

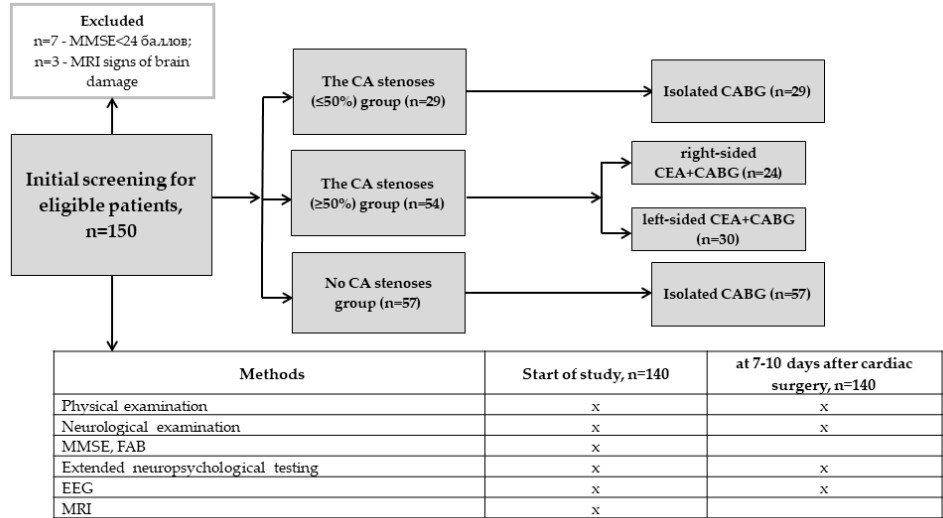

**Figure 1.** Overview of the study design.

The inclusion criteria were as follows: Aged between 45 and 74 years and elective isolated on-pump CABG or combined CABG and CEA. Only right-handed subjects were included in the study to avoid any influence on cognitive status and EEG data regarding the factor of laterality.

The exclusion criteria were the presence of pathological changes in the central nervous system, as indicated by the results of multi-layered spiral computed tomography; depressive symptoms, as identified by the Beck Depression Inventory (BDI-II) (sum scores $\geq$ 8); dementia, as indicated by the Mini-Mental State Examination (MMSE) (sum scores $\leq$ 24) and Frontal Assessment Battery (FAB) (sum scores $\leq$ 11); life-threatening arrhythmias; functional class IV heart failure, according to the New York Heart Association (FC NYHA IV) guidelines; chronic obstructive pulmonary disease; malignant pathology; diseases of the central nervous system; brain injury. Patients receiving anxiolytic therapy were also excluded from the study.

All patients underwent standardized physical, neurological, and instrumental examinations. The examiners were blind to the cognitive status of the patients. The severity of the coronary lesions was assessed using the findings of coronary angiography (Innova 3100; GE Medical Systems, Carrollton, TX, USA). Carotid artery ultrasound and echocardiography with estimation of the left ventricular ejection fraction (LVEF) were performed with the Vivid 7 ultrasound machine (GE Medical Systems).

The patients received baseline and symptomatic therapy before and after surgery, consistent with the general principles of treatment for the patients with CAD, chronic heart failure, and hypertension (National Recommendations, 2020) (see Table 1).

**Table 1.** The clinical and anamnestic characteristics of the patients before cardiac surgery (n = 140).

| Variable | Value |
|---|---|
| Age, years, Me (Q25; Q75) | 59 (56; 64) |
| Mini-mental state, scores, Me (Q25; Q75) | 27 (26; 28) |
| Frontal assessment battery, scores, Me (Q25; Q75) | 16 (15; 17) |
| BDI-II, scores, Me (Q25; Q75) | 3 (2; 4) |
| Educational attainment, years, n (%) | |
| 8–10 | 101 (72) |
| $\geq$15 | 39 (28) |
| Functional class of angina, n (%) | |
| I-II | 94 (67) |
| III | 46 (33) |
| Functional class NYHA, n (%) | |
| I-II | 109 (78) |
| III | 31 (22) |
| History of myocardial infarction, n (%) | 104 (74) |
| Fraction of left ventricle ejection, %, Me (Q25; Q75) | 58 (54; 62) |
| Type 2 of diabetes mellitus, n (%) | 48 (34) |
| Carotid arteries stenoses, n (%) | |
| One-sided $\leq$50% | 29 (21) |
| One-sided 70–99% | 7 (5) |
| Two-sided $\geq$50% | 47 (34) |
| History of stroke, n (%) | 15 (11) |
| Cardiopulmonary bypass time, min, Me (Q25; Q75) | 90 (83; 97) |
| Aorta cross-clamping time, min, Me (Q25; Q75) | 68 (56; 50) |
| Medication, n (%) | |
| ACEi | 124 (89%) |
| Statin | 94 (67%) |
| Beta-blockers | 137 (98%) |
| Antiplatelet drugs | 135 (96%) |
| CCB | 59 (42%) |
| Nitrates | 23 (16%) |

ACEi, angiotensin-converting enzyme inhibitor; CCB, calcium channel blockers; NYHA, heart failure by the New York Heart Association.

All surgical interventions in patients of the isolated CABG and CABG+CEA groups with the use of cardiopulmonary bypass, normothermia and 25–30% hemodilution were

carried out. In almost all cases, a blood pharmaco-cold cardioplegia was used. The standard anesthesia and infusion scheme was performed for all types of procedures. All stages of the surgery were accompanied by invasive hemodynamic control and real-time monitoring of cerebral cortex oxygenation (rSO2) (INVOS 3100; Somanetics, Troy, MI, USA). For simultaneous intervention (CABG+CEA), the initial stage of surgery was endarterectomy with arterial plasty and a xenopericardial patch.

### 2.2. Neurophysiological Assessment

The patients were assessed at baseline (1–3 days before surgery) and 7–10 days after surgery.

The cognitive screening and the extended neuropsychological test battery to evaluate three functional cognitive domains (psychomotor and executive function, attention and short-term memory) were conducted. Parallel test versions were used in repeated measurements in order to minimize learning effects. The neuropsychological test battery has been previously described [33,38]. Postoperative cognitive decline after CABG was determined by a 20% decrease in the cognitive score compared to baseline in 20% of the tests [31].

EEGs were recorded via a 62-channel Quik-cap (NeuroScan, El Paso, TX, USA). The scalp locations of the electrodes were based on the modified 10/10 System, and a nose bridge electrode was used as a reference. Bipolar eye movement electrodes were applied to the canthus and cheek bone to monitor eye movement artifacts. The EEGs were recorded using an NEUVO-64 system (NeuroScan, El Paso, TX, USA) in the eyes-closed and eyes-open conditions, in a dimly lit, soundproof, electrically shielded room, and recording lengths were about 10 min. The amplifier bandwidths were 1.0 to 50.0 Hz, and EEGs were digitized at 1000 Hz. The data were analyzed off-line using the Neuroscan 4.5 software program (Compumedics, TX, USA). We performed visual inspections for eye movements, electromyographic interferences, and other artifacts. Artifact-free EEG fragments were divided into 2 s epochs and underwent Fourier transformations. For each subject, the EEG power values were averaged within the theta1 (4–6 Hz), theta2 (6–8 Hz), alpha1 (8–10 Hz), and alpha2 (10–13 Hz) ranges [39]. The EEG power values of each channel for every subject in each band were obtained. The next step was the clustering of data recorded in 56 leads into 5 electrode zones symmetrically in the left and right hemispheres: frontal, central, parietal, occipital and temporal. The midline sites (Fpz, Fz, etc.) were excluded. The clustering of nearby electrodes was conducted to increase statistical significance.

### 2.3. Statistical Analysis

All data were analyzed using STATISTICA 10.0 (StatSoft, Tulsa, OK, USA). The normality of the distribution of clinical and demographic parameters was tested using the Kolmogorov–Smirnov test. Most of the clinical parameters as well as cognitive indicators were not normally distributed and were analyzed using the Wilcoxon and Mann–Whitney tests. EEG data were normalized using the logarithm transformation and further analysis of the EEG data was carried out using a repeated-measures ANOVA. Levene's test was used to assess the equality of variances for EEG variables. The Greenhouse–Geisser correction of statistical significance was used in ANOVA. Post hoc pairwise comparisons for groups of patients were performed using Newman–Keuls multiple comparison tests.

## 3. Results

### 3.1. The Effect of Small Stenoses CA (≤50%) on the Postoperative Neurophysiological Changes in On-Pump CABG Patients

#### 3.1.1. Neurophysiological Data

This analysis included 86 patients who had undergone isolated CABG. According to the results of the preoperative examination, they were divided into two groups: those with CA stenoses of less than 50% (n = 29) and those without stenoses (n = 57).

The postoperative period was standard in all the patients, without adverse cardiovascular events (intraoperative and postoperative heart attacks, strokes, life-threatening arrhythmias, bleeding, etc.).

POCD occurred in 22 (76.0%) patients with CA stenoses and in 32 (61%) patients without stenoses after isolated CABG (OR = 1.99, 95% CI = 0.77–5.18, Z = 1,42, $p$ = 0,15). Thus, the incidence of POCD had a tendency of an increasing number of cases in the CA stenoses group.

The POCD structure consisted of a decrease in the psychomotor and executive function, as well as short-term memory in both groups. At the same time, the patients with CA stenoses made more errors in the tests of executive functions ($p \leq 0.05$), and patients without stenoses had more missed signals in the same tests. In the domain of short-term memory, between-group differences were obtained in the 10-nonsense-syllable memorizing test ($p$ = 0.04).

### 3.1.2. EEG Data

For the next stage of the analysis, a repeated-measures ANOVA with a between-subjects factor of GROUP (two levels: with CA stenoses of less than 50%/without stenoses), and within-subjects factors of EXAMINATION TIME (two levels: before/after surgery), AREA (five levels: frontal, central, parietal, occipital and temporal), and LATERALITY (two levels: left/right hemisphere) was conducted. The significant factors and interactions associated with the GROUP factor are found in the theta2, alpha1 and alpha2 EEG ranges.

The statistically significant interactions of the factors GROUP × EXAMINATION TIME ($F_{1.84}$ = 4.95, $p$ = 0.03) and GROUP × EXAMINATION TIME × AREA × LATERALITY ($F_{4.336}$ = 3.54, $p$ = 0.02) were found in the theta2 range of EEG resting state with eyes closed. The patients with CA stenoses had higher values of the theta2-rhythm power at 7–10 days after CABG in comparison to the patients without stenoses (Figure 2). In addition, the CA stenoses group had higher values of rhythm power in the left hemisphere in the frontal and centroparietal cortical regions and in the right hemisphere in all sites, except for the occipital regions.

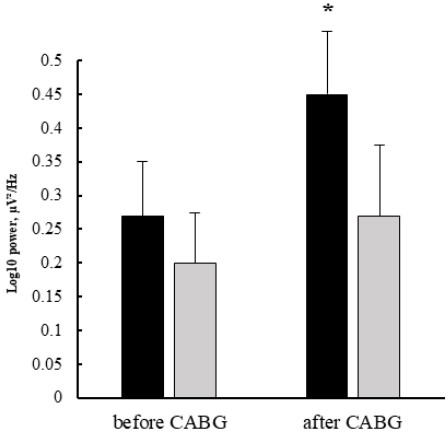

**Figure 2.** Differences in the theta2-rhythm power of EEG resting state with eyes closed in patients who underwent on-pump CABG, depending on the presence of CA stenoses less than 50%: dark columns—the patients with CA stenoses, light columns—the patients without CA stenoses, error bars denote SE, *—$p$ < 0.05 Newman–Keuls multiple comparison test.

The significance of the GROUP factor was obtained in the theta2, alpha1 and alpha2 frequency ranges of EEG resting state with eyes open ($F_{1,84}$ = 4.68, $p$ = 0.034; $F_{1,84}$ = 3.88, $p$ = 0.05 and $F_{1,84}$ = 4.96, $p$ = 0.029, respectively). The patients with CA stenoses had higher power values of these rhythms compared to patients without stenoses before and after cardiac surgery.

Additionally, the analysis of EEG resting state with open eyes revealed a statistically significant interaction of the factors GROUP × EXAMINATION TIME × AREA × LATERALITY ($F_{4,336} = 2.77$, $p = 0.04$) in the alpha2 frequency range. Before surgery, the power of rhythm was higher in the right frontal ($p = 0.04$) and central ($p = 0.025$) areas in patients with CA stenoses compared to patients without stenoses. There were no between-group differences in the left hemisphere. After CABG, the patients with CA stenoses had higher power values in the frontal ($p = 0.03$ and $p = 0.036$, respectively), central ($p = 0.007$ and $p = 0.01$, respectively) and parietal ($p = 0.02$ and $p = 0.019$, respectively) regions of the left and right hemispheres (Figure 3).

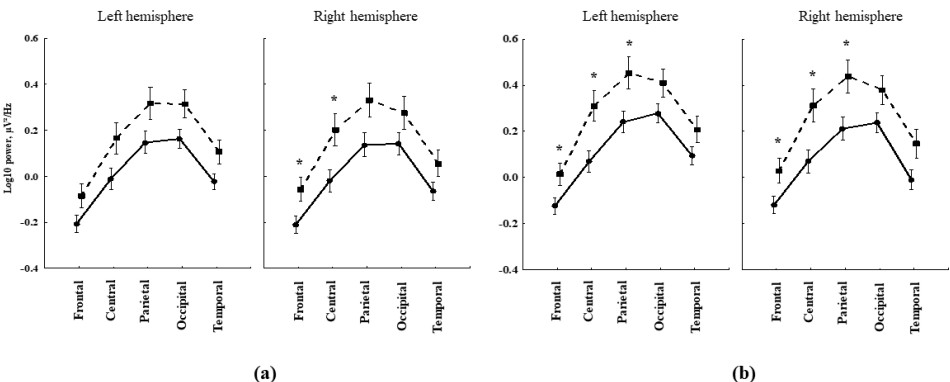

(a)　　　　　　　　　　　　　　　　　　　　(b)

**Figure 3.** Lateral differences in the alpha2-rhythm power changes of EEG resting state with eyes open in patients after on-pump CABG, depending on the presence of CA stenoses less than 50%: (**a**) before cardiac surgery; (**b**) after surgery; solid lines—the patients without stenoses, dashed lines—the patients with CA stenoses, error bars denote SE, \*—$p < 0.05$ Newman–Keuls multiple comparison test.

Thus, the presence of CA stenoses of less than 50% in patients who underwent on-pump CABG was associated with more pronounced signs of EEG of brain dysfunction. Both before and after CABG, the patients with CA stenoses demonstrated higher values of theta2- and alpha-rhythm power compared to the patients without CA stenoses.

*3.2. The Postoperative Neurophysiological Status Changes in the Patients after Combined On-Pump CABG and CEA*

3.2.1. Neurophysiological Data

This analysis included 111 patients who have undergone combined coronary and carotid artery revascularization or isolated CABG. According to the results of the preoperative examination, they were divided into three groups: the group of left-sided CEA+CABG (n = 30), the group of right-sided CEA+CABG (n = 24), and the group of isolated CABG (n = 57). The patients with combined coronary and carotid surgery had significant CA stenoses as assessed by digital angiography (NASCET criteria).

No adverse cardiovascular events (myocardial infarction, stroke, death, and repeated unplanned revascularization) were observed in the patients in the early postoperative period for simultaneous CABG+CEA or isolated CABG. In this cohort, POCD occurred in 34 (63.0%) patients with CABG+CEA, and in 32 (61%) patients with isolated CABG (OR = 1.33, 95% CI = 0.62–2.84, $p = 0.59$). Significant between-group differences were detected for the psychomotor and executive function indicators. At 7–10 days after surgery, the psychomotor speed in two neurodynamic tests was higher in the CABG group than in the group with CABG+CEA ($p = 0.0002$ and $p = 0.005$, respectively). In addition, the CABG patients had better indicators of executive control in the same tests at 7–10 days after surgery compared to the patients with CABG+CEA ($p = 0.0004$ and $p = 0.02$, respectively).

### 3.2.2. EEG Data

A repeated-measures ANOVA with a between-subjects factor of GROUP (three levels: CABG+left-sided CEA/CABG+right-sided CEA/isolated CABG) and within-subjects factors of EXAMINATION TIME (two levels: before/after surgery), AREA (five levels: frontal, central, parietal, occipital and temporal), and LATERALITY (two levels: left/right hemisphere) was conducted. The significant factors and interactions associated with the GROUP factor are found in EEG resting state with eyes closed in the theta1 frequency range.

There was a significant factor in EXAMINATION TIME—$F_{1.108} = 46.6$, $p \leq 0.0001$. It was found that the theta1 power increased after surgery at 7–10 days of the postoperative period as compared with the preoperative level both in the CABG patients and in the two CABG+CEA groups. This effect was more pronounced in CABG+right-sided CEA patients ($p = 0.0001$); they differed also from the isolated CABG group at 7–10 days after surgery ($p = 0.026$) (Figure 4).

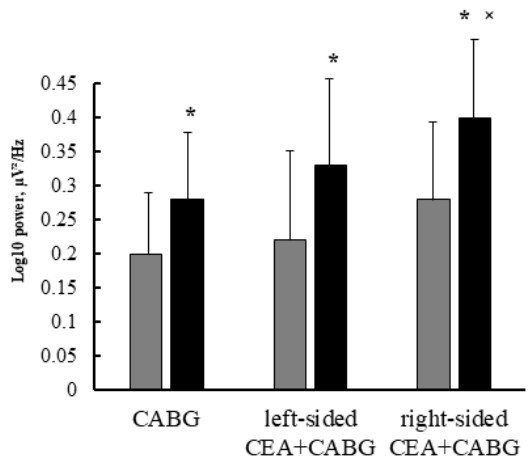

**Figure 4.** The postoperative theta1 rhythm power changes in the patients undergoing isolated CABG and simultaneous intervention (CABG+CEA): grey columns—the CABG patients, dark columns—the CABG+CEA patients; error bars denote SE, *—$p < 0.05$ Newman–Keuls multiple comparison test for the postoperative indicators in comparison to preoperative levels, ×—$p < 0.05$ Newman–Keuls multiple comparison test for the postoperative indicators in CABG+CEA group as compared to CABG group.

The interaction of factors GROUP × LATERALITY ($F_{2,108} = 3.22$, $p = 0.04$) was also significant. The left-sided CEA+CABG patients demonstrated the fewest lateral differences of theta1 power. The isolated CABG and CABG+right-sided CEA patients had higher theta1 power values in the left hemisphere as compared to the right one. This effect was more pronounced in CABG+right-sided CEA patients ($p = 0.0004$ and $p = 0.00008$, respectively) (Figure 4).

Another significant interaction of factors GROUP × EXAMINATION TIME × AREA × LATERALITY ($F_{8,432} = 2.15$, $p = 0.048$) was revealed. The theta1 power differences between the patients who underwent isolated CABG and right-sided CEA+CABG were found. Before surgery, the right-sided CEA+CABG patients had higher theta1 power values than isolated CABG patients only in the frontal cortical regions in both hemispheres ($p = 0.001$ and $p = 0.047$, respectively). After surgery, the between-group differences were more pronounced in the left hemisphere. The right-sided CEA+CABG patients had higher theta1 power values than isolated CABG patients in all cortical regions, except occipital. In the right hemisphere, the between-group differences were only in the frontal, central and temporal regions, as seen in Figure 5.

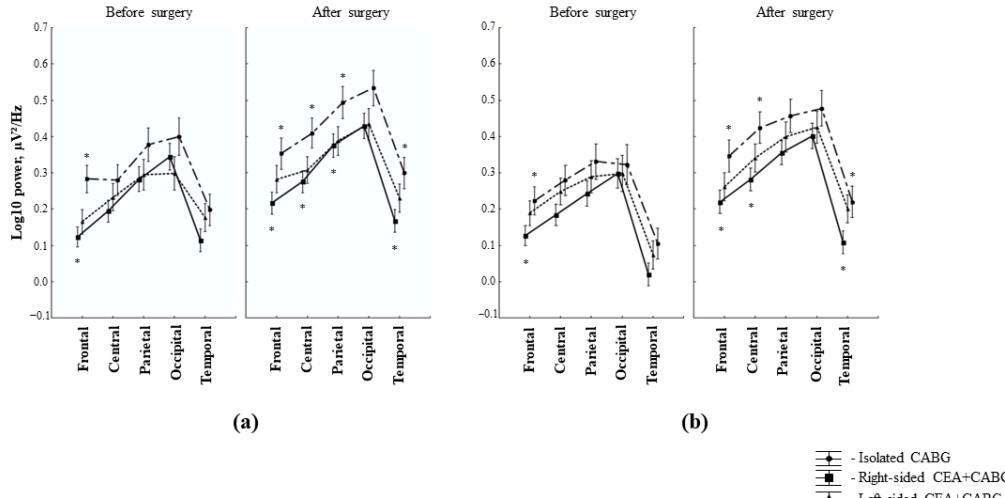

**Figure 5.** The topography of the postoperative theta1 rhythm power changes in the patients undergoing isolated CABG and simultaneous intervention (CABG+CEA): (**a**) left hemisphere; (**b**) right hemisphere; error bars denote SE, *—$p < 0.05$ Newman–Keuls multiple comparison test for between-group differences.

## 4. Discussion

As found in our study, the frequency of POCD was higher in patients with less than 50% CA stenoses in comparison to the patients without them (76% vs. 61%). However, the frequency was comparable in combined coronary and carotid surgery and isolated CABG (63% vs. 61%). The POCD structure both in the patients with CA stenosis of less than 50% and in patients with hemodynamically significant CA stenoses (70–99%) consisted of executive function decline, which was determined as the non-successful performance of neurodynamic tasks in these patients. Previously, it has been shown that for the correct assessment of the signal sequence in neurodynamic tests, a high level of indicative activity is required. This causes increased brain energy consumption [40]. It can be assumed that highly organized cognitive activity is disrupted by the deterioration of cerebral blood flow in patients with CA stenoses.

We also demonstrated that the patients with CA stenosis of less than 50% had more pronounced signs of brain dysfunction as compared with patients without stenoses. These changes were diffuse and expressed as higher power values of resting state EEG in the frequency band from 6 to 13 Hz. Earlier, it has been shown that an increase in the slow rhythm power is associated with a decrease in the level of cortical activation and may be a reflection of chronic cerebral ischemia [41,42]. It should be noted that these pathological EEG signs were observed in patients with CA stenosis of less than 50% already in the preoperative period and persisting after surgery. One of the possible causes of neurological complications in patients with hemodynamically insignificant CA stenoses may be the instability of small atherosclerotic plaques with the development of vasoconstrictor and procoagulant effects [43]. There is an assumption that the atherosclerosis in patients with multiple vascular lesions may proceed more aggressively [44]. We may propose that such patients probably develop a more pronounced systemic inflammatory response associated with cardiopulmonary bypass. Earlier experiments showed that the combined effect of ischemia and hypoxia induces an increase in the production of pro-inflammatory cytokines (TNF-α, IL-1β and IL-6) in the brain, which contributes to damage and increased permeability of the blood–brain barrier, and as a consequence, the development of brain edema [45,46]. In addition, cerebral blood flow autoregulation may be disrupted more often in patients with CA stenoses, leading to the decrease in the brain's resistance to acute ischemia and hypoperfusion associated with cardiopulmonary bypass [36,47]. The state of the circle of Willis and the density of leptomeningeal collaterals also contribute to brain

hemodynamic parameters [48,49]. On the other hand, the interaction between macro- and microcirculation requires attention in regard to postoperative neurophysiological changes in the patients after cardiac surgery. Earlier, it has been found that carotid atherosclerosis, white matter hyperintensities and lacunar infarction are associated with and commonly contribute to the deterioration of neurological function [50,51].

Additionally, one conclusion we reached was that patients with CA stenoses of less than 50% are vulnerable to the effects of the factors that accompany cardiac surgery using cardiopulmonary bypass compared with patients without CA lesions. The presence of even hemodynamically insignificant stenoses in cardiac surgery patients makes it possible to include them in the group at increased risk of brain damage in the perioperative period. This category of patients should be considered as requiring more careful preoperative management, the use of methods of perioperative protection of the brain, the choice of safe strategies for myocardial revascularization and the involvement of methods of cognitive rehabilitation.

A next finding of our study was that the patients who underwent right-sided CABG+CEA are characterized by the most pronounced theta power changes and generalized "slow-down" of the EEG compared with patients who underwent isolated CABG and left-sided CABG+CEA.

It has been recently reported that severe carotid stenosis can disturb the hemodynamic balance, illustrated by blood flow laterality [52]. As shown by the results in our work, a contralateral stenosis of the CA was observed in 86% of cases in patients who underwent CABG+right-sided CEA. Our study showed that the right hemisphere was more vulnerable intraoperatively. In the study by M. Hedberg and K.G. Engström [53], it was shown that a stroke occurs more often in the right than in the left hemisphere in the early postoperative period of cardiac surgery.

Therefore, the results of the study lead us to conclude that on-pump cardiac surgery is a traumatic brain event, regardless of the type of intervention. Bilateral CA lesion increases the severity of cortical dysfunction in the postoperative period, which requires the use of complex brain protection methods. At the same time, it is worth noting that combined CABG and CEA surgery in comparison with isolated CABG does not lead to more significant brain damage. This fact is an additional argument that makes the strategy of one-stage revascularization of the brain and heart justified.

A set of characteristics of the resting EEG, including a postoperative theta power increase and generalized "slowdown", was obtained in our study. This is a universal brain response to damage, indicating an imbalance between cortical and subcortical structures and a decrease in the functional activity of the cerebral cortex [8,22,23,41,42]. The topography of postoperative EEG activity disturbances included the frontal, temporal and parieto-occipital regions. It is assumed that patients with cardiovascular diseases are most susceptible to ischemic changes in the frontal regions of the brain, which plays a key role in the executive function, action planning and working memory [3–5,40]. At the same time, neurodegenerative brain damage, first of all, is detected in the hippocampus and adjacent areas of the brain (cingulate and temporo-parietal cortex) [23,27,28]. Recent studies of cognitive disorders in a cohort of cardiovascular disease patients have shown that it is difficult to differentiate neurodegenerative and ischemic patterns of brain damage; to a greater extent, researchers are inclined to a mixed etiology of cognitive deficits associated with both the progression of atherosclerotic changes in brain vessels and age-related neurodegenerative changes [7,8,54].

## 5. Conclusions

The high frequency of cognitive decline in the postoperative period in patients who underwent cardiac surgery with the use of cardiopulmonary bypass and the ambiguity of the mechanisms underlying the development of brain damage encourage further study of this phenomenon in a cohort of patients with cardiovascular diseases. Our results show that an integrated approach using modern methods of neuropsychological testing and

computerized EEG allows for timely diagnosis of postoperative cognitive disorders and can be useful in determining the effectiveness and safety of cardiac surgery. We showed that cardiac surgical interventions with cardiopulmonary bypass are associated with a high risk of episodes of brain ischemia. This may be accompanied by specific rearrangements of frequency–spatial patterns of electrical brain activity, dependent on the degree of damage to coronary and carotid arteries. The information obtained can be used to optimize the process of preoperative management and the search for anesthesiologic brain protection and safe surgical techniques and strategies for myocardial revascularization, as well as postoperative rehabilitation of this category of patients.

**Author Contributions:** Conceptualization, I.T., O.T. and O.B.; methodology, I.T. and O.T.; validation, I.T. and O.T.; formal analysis, I.T. and D.K.; investigation, D.K., I.K., I.S. and A.S.; data curation, D.K., I.K., I.S., A.S. and O.M.; writing—original draft preparation, I.T.; writing—review and editing, O.T. and O.B.; project administration, O.T.; funding acquisition, O.B. All authors have read and agreed to the published version of the manuscript.

**Funding:** The authors declare that this study received funding from the Federal State Ministry of Science and Education of Russian Federation (The fundamental theme No. 122012000364-5 dated 20 January 2022). The funder was not involved in the study design, collection, analysis, interpretation of data, the writing of this article or the decision to submit it for publication.

**Institutional Review Board Statement:** The study was conducted in accordance with the Declaration of Helsinki, and approved by the Institutional Ethics Committee of the Research Institute for Complex Issues of Cardiovascular Diseases (01/2011-2520). The study was registered on the ClinicalTrials.gov (NCT05172362).

**Informed Consent Statement:** Informed consent was obtained from all subjects involved in the study.

**Data Availability Statement:** Not applicable.

**Conflicts of Interest:** The authors declare no conflict of interest. The funder had no role in the design of the study; in the collection, analyses, or interpretation of data; in the writing of the manuscript; or in the decision to publish the results.

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
