# Peer review of "Effect of Carotid Stenosis Severity on Patterns of Brain Activity in Patients after Cardiac Surgery"

_applsci, doi:10.3390/app13010020_

Round 1
Reviewer 1 Report
In this paper, the authors investigated the effect of carotid stenosis severity on patterns of brain activity in patients after cardiac surgery using EEG analysis. The topic is interesting. Overall, the paper is well structured. The results provide reference for pathophysiological research and clinical applications. However, some improvements are essential to meet publishable standard.
1. The acronym should be used when it has been explained, e.g., cognitive impairment.
2. In the patients with a stroke history, did you record the affected region? This may influence the observation results.
3. Please provide the sampling frequency and resolution of the EEG device.
4. Did you test the homogeneity of variance before performing ANOVA?
5. Lines 195-196: Please simplify the subtitle if possible.
6. Lines 198-205: The information is aforementioned. Should be integrated into the introduction. I suggest to remove the whole subsection 3.1.1.
7. Instead, you may need a subsection to introduce the details of subjects in Table 1.
8. The authors observed that the right hemisphere was more vulnerable intraoperatively. In the 36 patients with one-sided carotid atherosclerosis, did you observe any different in the EEG-derived parameters between hemispheres?
9. Lines 237-238: In figure 1, the difference between two groups shows the same trend before and after the CABG. The comparison in CABG-related difference (i.e., between pre- and post- CABG values) between two groups may be more valuable.
10. Regarding the hemodynamic effect of carotid arteries on cerebral blood flow, the role of autoregulation and collateral circulation deserves more attention. For example, the circle of Willis, which is a major mechanism of cerebral collateral circulation, has an intact structure in less than 50% of general population (Refer: 10.1109/ACCESS.2020.3007737). The density of leptomeningeal collaterals is also highly patient-specific (Refer: 10.1177/0271678X18805209). These factors were not included in this pilot study, but need to be considered in future studies to develop a roadmap for comprehensive evaluation of the hemodynamic influence of carotid plaques and CEA.
11. The interaction between macro- and microcirculation deserves further discussion. As the authors mentioned in lines 368-369, a possible cause of neurological complications in patients with hemodynamically insignificant CA stenoses might be the instability of small atherosclerotic plaques. In some parallel studies, it was observed that carotid atherosclerosis (CAS), white matter hyperintensities (WMHs) and lacunar infarction (LI) are associated and commonly contributed to the deterioration of neurological function (Refer: 10.1007/s00415-021-10899-7, 10.3389/fneur.2020.00214). The hemodynamic risks on both macro- and microvascular levels, and their interaction with structural changes in the brain, need to be discussed.
12. There are some minor errors in language and format. I would like to advise the authors to find a native English speaker for proofreading.
Author Response
Dear Reviewer,
Thank you very much for your comments.
Please see below for answers and comments to your questions.
- Question 1: We have applied the acronym “CI” where necessary. We apologize for the inaccuracy.
- Question 2: Unfortunately, we didn’t record the affected region in the patients with a stroke history. We agree with the honorable reviewer that this could affect the interpretation of the results. It should be noted that, the previous stroke had no clinical manifestations (motor or sensory deficit) in the patients at the time of the study. We would like to take this into account in our future research.
- Question 3: The amplifier bandwidths were 1.0 to 50.0 Hz, and EEGs were digitized at 1000 Hz. This is indicated in the text (lines 196-197).
- Question 4: EEG data were normalized using the logarithm transformation. Levene's test used to assess the equality of variances for EEG variables.
- Question 5. We simplified the subtitle 3.1 as “The effect of small stenoses CA (≤50%) on the postoperative neurophysiological changes in on-pump CABG patients”.
- Question 6. The subsection 3.1.1 was deleted by transferring this information to the Introduction section.
- Question 7. The subsection 2.1. Subjects provide the information about our patients.
- Question 8. Table 1 provide the information of the overall sample of 140 cardiac surgery patients. Only 29 patients had one-sided carotid arteries stenosis ≤50%. Another 7 patients had one-sided carotid arteries stenosis 70-99%. We corrected the information in Table 1.
- Question 9. The CABG-related differences (between pre- and post- CABG values) between two groups of patients with and without stenoses CA ≤50% demonstrated the statistically significant interactions of the factors GROUP × EXAMINATION TIME (F 84=4.95, p=0.03). Before CABG the patients with CA stenoses had only trend to higher values of the theta2-rhythm power. But at 7–10 days after CABG the between-group difference had the statistical significance (p<0,05).
- Question 10. We also believe that the role of hemodynamic effect of the circle of Willis and leptomeningeal collaterals is significant. We will be considered the effect of these factors in future studies.
- Question 11. We have included the proposed articles in the Discussion section. We are very appreciative of the reviewer's comments. We were able to improve our manuscript, based on these criticisms.
- Question 12. The manuscript also was submitted to professional translation service to revise from a formal language point of view.

Reviewer 2 Report
This study investigates the effect of carotid stenosis severity on brain activity in patients after cardiac surgery. The manuscript is written clearly, but there are minor drawbacks:
1. In the introduction it should be pointed out the novelty of the manuscript compared to the state-of-art literature.
2. In Figure 2, the measurement unit on the y-axis should be correctly specified.
Author Response
Dear Reviewer,
Thank you very much for your comments.
Please see below for answers and comments to your questions.
- Question 1: We corrected the Introduction section to highlight the novelty of our manuscript compared to the state-of-art literature. We would like to emphasize that this high-resolution EEG study demonstrated the brain activity changes associated with postoperative cognitive decline in patients with different severity of carotid arteries stenos after isolated and combined cardiac surgery. We found no similar studies in the research databases.
- Question 2: We corrected the measurement unit on the y-axis in Figure 2. We apologize for the inaccuracy. Because EEG data were normalized using the logarithm transformation, we indicate the measurement unit on the y-axis as Log10 power, µV²/Hz.
- The manuscript also was submitted to professional translation service to revise from a formal language point of view.

Reviewer 3 Report
On pump cardiac surgery is assoaciated with signficant comorbity and mortality .Therefore,the researchers must justify the sample size ,provide a flow chart to depict the patient status from inclusion to discharge ,so the actual number of patients studied can be known .The authors must mention appropriate statistical method in the captions of tables and figure.The number of patients studied are different at different places of manuscript ,please unify
Author Response
Dear Reviewer,
We are very appreciative of the reviewer's comments. We were able to improve our manuscript, based on these criticisms.
Please see below for answers and comments to your questions.
Therefore, the researchers must justify the sample size, provide a flow chart to depict the patient status from inclusion to discharge, so the actual number of patients studied can be known. The authors must mention appropriate statistical method in the captions of tables and figure. The number of patients studied are different at different places of manuscript, please unify.
- According to the comment of the honorable reviewer about the sample size, we provide the additional figure (depicted as Figure 1) which presented the study design overview. So, we included 150 eligible patients according to inclusion or exclusion criteria, but ten patients were excluded after MRI scanning and cognitive screening. Twenty-nine patients had one-sided carotid arteries stenosis ≤50% and 57 patients had no carotid stenoses, these 86 patients underwent isolated cardiac surgery. Another 7 patients had one-sided carotid arteries stenosis 70-99%, two-sided ≥50% stenoses of carotid arteries had 47 patients, these 54 patients underwent combined cardiac surgery. We corrected also the information in Table 1.
- We mentioned appropriate statistical methods in the captions of figures. Table 1 provide the information of the overall sample. There are no statistical comparisons.
- The manuscript also was submitted to professional translation service to revise from a formal language point of view.

Round 2
Reviewer 1 Report
My earlier comments have been well addressed. Despite the limitations, I suggest the publication of this paper considering the methodological innovation and its potential for future research. Please proofread and double check the format.